# NeuralEQ: Neural-Network-Based Equalizer for High-Speed Wireline Communication

## Abstract

Massive data processing is required by various applications such as ML, video streaming, cloud services, etc. In such systems, I/O bandwidth must be scaled up to prevent any performance degradation due to the limited data transfer rates. To meet this demand, wireline communication recently started adopting PAM4 signaling and DSP-based equalizers. However, multi-level signaling and conventional equalizing techniques degrade the bit-error-rate (BER) performance significantly. To mitigate this problem, this paper proposes a novel neural network architecture that mimics the forward-backward algorithm estimating the posterior probabilities in Hidden Markov Models. The proposed neural network overcomes the existing equalizer performance, such as feed-forward equalizers or decision-feedback equalizers, while reducing the complexity of the forward-backward algorithm.

## 1 Introduction

Recent advances in ML/AI technologies have benefited our daily life by enabling services like searching, recommendation, and translation. The rapid growth of ML applications, increasing the amount of data and computation, has driven enormous demands on high-performance computing systems. Accordingly, the I/O bandwidth and computing power of such systems must be scaled up to support large data transactions between computing cores without experiencing performance degradation due to their limited data transfer rates. To meet such high bandwidth demand for high-performance computing systems, the PCIe standard is being developed up to 64 Gbps for the next generation, and data-rate of NVlink specializing in GPU communication reaches 50 Gbps (Temuçin et al., 2021). Data-rate of the ethernet protocol is also being developed up to 112 Gbps to handle the high network load.

Although demands for higher data-rate for wireline communication keep increasing, limited bandwidth of wireline channels poses problems in data transmission. Different from wireless communication, wireline channels are time-invariant. In addition, pursuing low latency and good I/O energy efficiency, wireline communication has relied on simple modulation like PAM2 and simple equalizers, followed by symbol-by-symbol detection. However, data-rate is increasing so rapidly even a short channel causes severe signal distortion, making it difficult to restore data at the receiver. To increase the data-rate further, recently, wireline communication started adopting multi-level signaling like PAM4 and DSP-based equalizers along with analog-to-digital converters (ADCs) on the receive side. Although PAM4 signaling enables data-rate to reach over 100 Gbps, as (Cordova et al., 2021; de Abreu Farias Neto et al., 2020; Krupnik et al., 2020), due to the peak power constraint, PAM4 degrades SNR and makes the signal more vulnerable to inter-symbol interference (ISI) caused by limited channel bandwidth, resulting in bit-error-rate (BER) degradation. Under these circumstances, equalizers to compensate for the channel loss play a critical role in high-speed wireline communication.

Equalizers can be primarily divided into a linear equalizer, e.g., feed-forward equalizer (FFE), and a nonlinear equalizer, e.g., decision-feedback equalizers (DFE). FFE is a discrete-time finite-impulse-response (FIR) filter that boosts the input in the frequency range where channel loss is high. It is simple to implement, but since incoming noise is also amplified, symbol detection after FFE shows limited BER performance. On the other hand, DFE does not boost noise and shows better BER, but the timing constraint of the feedback loop makes it difficult to design at a high speed.

The loop-unrolling, or speculative, technique mitigates the timing constraint but increases hardware complexity significantly, especially in PAM4, resulting in a loss of energy efficiency. Moreover, DFE alone cannot remove the pre-cursor ISI. Therefore, high-speed wireline communication typically uses both FFE and DFE to improve BER (Dikhaminjia et al., 2018).

Recently, thanks to the development of deep learning, receivers using neural networks, especially recurrent neural networks (RNNs), have been studied, (Ye et al., 2018; Zhou et al., 2019; Kechriotis et al., 1994; Kim et al., 2020; Gomez Diaz et al., 2022), showing excellent performance for sequence detection. However, for wireline communication, RNNs also have disadvantages that 1) high-speed implementation is difficult due to the timing constraint (past computation results are required for the current operation), and 2) sequence detection incurs large latency.

In this paper, we propose a novel neural network architecture that is amenable to high-speed implementation and shows better BER than existing equalizers. The proposed neural network employs feed-forward architecture without any data-dependency-induced timing constraints, which allows efficiently pipelined hardware architecture for high-speed receivers. For better BER performance, the proposed network is trained to mimic the forward-backward (FB) algorithm that has been used to estimate posterior probabilities of the states in Hidden Markov Models (HMMs), while reducing the computational complexity of the FB algorithm.

This paper makes the following contributions:

- Inspired by the forward-backward algorithm, a novel neural network, NeuralEQ, whose structure is similar to that of the forward-backward algorithm is proposed.
- We verify that the computational complexity of the proposed NeuralEQ is lower than that of the forward-backward algorithm. While having much lower complexity, NeuralEQ shows superior BER performance compared to conventional equalizers.
- More than half the weights of NeuralEQ can be pruned without performance degradation, which reduces the complexity significantly. Moreover, NeuralEQ is found to be robust against ISI variation.

## 2 BACKGROUND

The core idea of this paper is to develop a novel neural network architecture that mimics the forward-backward (FB) algorithm in order to overcome the existing equalizer (EQ) performance such as FFE and DFE while reducing the complexity of the FB algorithm. In the following, a brief description of the FB algorithm and how it can be applied to wireline communication will be provided.

### 2.1 FORWARD-BACKWARD ALGORITHM

The FB algorithm, which is also known as BCJR (Bahl et al., 1974) or MAP decoder, is widely used in modern wireless communication with its excellent performance. However, since wireline communication requires a much higher data rate, it is challenging to run a computationally heavy algorithm such as FB. It would be helpful to look at the formula in detail to understand the complexity of the FB algorithm.

The FB algorithm infers the maximum posterior of all hidden states in a hidden Markov model. Let $S^t$ be a hidden state variable, which is an element of a set of hidden states $\mathbb{S}$, and $X^{1:t}$ be an observed sequence from time $1$ to $t$. Then the FB algorithm is summarized as follows.

$$\gamma^t = P(S^t, X^{1:t})P(X^{t+1:T}|S^t), \ 1 \leq t \leq T \tag{1}$$

$P(S^t, X^{1:t})$ is a probability of hidden states with a sequence from time 1 to t, which refers to forward probability. And $P(X^{t+1:T}|S^t)$ is a conditional probability of sequence occurs when the state at time t is given, which refers to backward probability. Each forward and backward probability can be computed using dynamic programming with the following equations. Note that $P(S^t = s_i, X^{1:t} = x^{1:t})$ is replaced with $\alpha_i^t$, $P(X^{t+1:T} = x^{t+1:T}|S^t = s_i)$ is replaced with $\beta_i^t$ for simplicity and $a_{ji}$ is the transition probability from j-th state to i-th state.

$$\alpha_i^t = \sum_{j \in |\mathbb{S}|} \alpha_j^{t-1} a_{ji} P(X^t = x^t | S^t = s_i), \ \ \beta_i^{t-1} = \sum_{j \in |\mathbb{S}|} \beta_j^t a_{ji} P(X^t = x^t | S^t = s_i) \tag{2}$$

Finally, element-wise product of forward and backward probability gives the posterior probability for a given sequence from 1 to T, which is expressed with $\gamma^t$ in Equation 1.

## 2.2 APPLYING FORWARD-BACKWARD ALGORITHM TO WIRELINE COMMUNICATION WITH LOSSY CHANNEL

The FB algorithm can be applied when wireline communication is modeled as HMM. To this end, four parameters in HMM should be defined: (1) the observed sequence, (2) the hidden states, (3) the transition probability, and (4) the conditional probability of observables given each hidden state. First, the observed sequence is defined as the channel output distorted by ISI. Typically wireline links are tested with random data patterns, so the channel input is assumed to be a random sequence without any coding (e.g. convolutional code). Note that, due to the channel ISI, correlation between adjacent channel outputs becomes nonzero. Second, each hidden state $s_i$ is defined as the possible channel input sequence with the length of the number of ISI, or $|\mathbf{ISI}|$, which determines the channel output without any noise. For example, in PAM4 signaling, there are $4^{|\mathbf{ISI}|}$ possible hidden states, each corresponding to a vector $\mathbf{z}_i \in (-1, -1/3, 1/3, 1)^{|\mathbf{ISI}|}$. Third, each hidden state has four possible state transitions with equal probability $1/4$. Finally, assuming an additive white Gaussian noise channel, the conditional probability distribution of the observable can be written as:

$$P(X^t = x | S^t = s_i) = \frac{1}{\sqrt{2\pi\sigma^2}} \exp[-\frac{1}{2}(\frac{x - \mu_i}{\sigma})^2], \text{where } \mu_i = (\sum_{j=1}^{T} \mathbf{z}_i^{T-j} \mathbf{h}^j) \tag{3}$$

Note that $\mathbf{h}$ is ISI coefficients. Figure 1a shows how PAM2 wireline communication can be modeled with HMM describing four parameters of HMM in wireline communication.

Exploiting the FB algorithm in wireline communication results in better BER performance compared to the conventional EQs such as FFE and DFE as demonstrated in Figure 1b. In this simulation, PAM4 signaling is used with $\mathbf{ISI}$=[1.0, 0.4, 0.2, 0.1], while the number of FFE and DFE taps are chosen 8 and 3, respectively, which are sufficient to compensate for ISI. Although the performance of the FB algorithm is excellent, its computational complexity is too high. There have been efforts to reduce the complexity for wireless communication (Wang et al., 2006; Thangarajah et al., 2011; Talakoub et al., 2007), but it is still difficult to use it in high-speed wireline communication. The number of hidden states increases exponentially with the length of ISI, and the forward and backward computations increase proportionally to the number of hidden states. Recently, the speed of wireline communication has reached 100 Gbps, so it is very challenging to implement the FB algorithm to operate at this speed, and power consumption is very high even if implemented. In this study, using a neural network, while reducing the complexity of the FB algorithm, an equalizer superior to conventional EQs in terms of BER performance is developed.

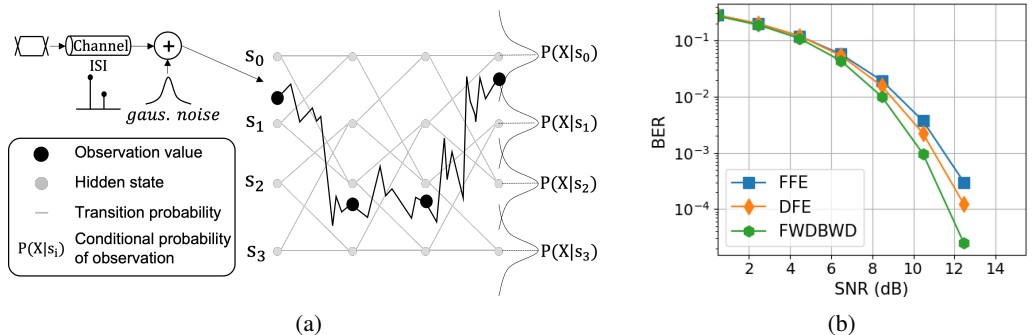

(a)  (b)

Figure 1: (a) Applying HMM to wireline communication, (b) Performance comparison between FFE, DFE, and FB

## 3 PROPOSED NEURAL EQUALIZER

There would be many ways to create an equalizer using the neural network. The first method would be to use the fully connected structure (Zhou et al., 2020; O'Shea & Hoydis, 2017; Schaedler et al.,

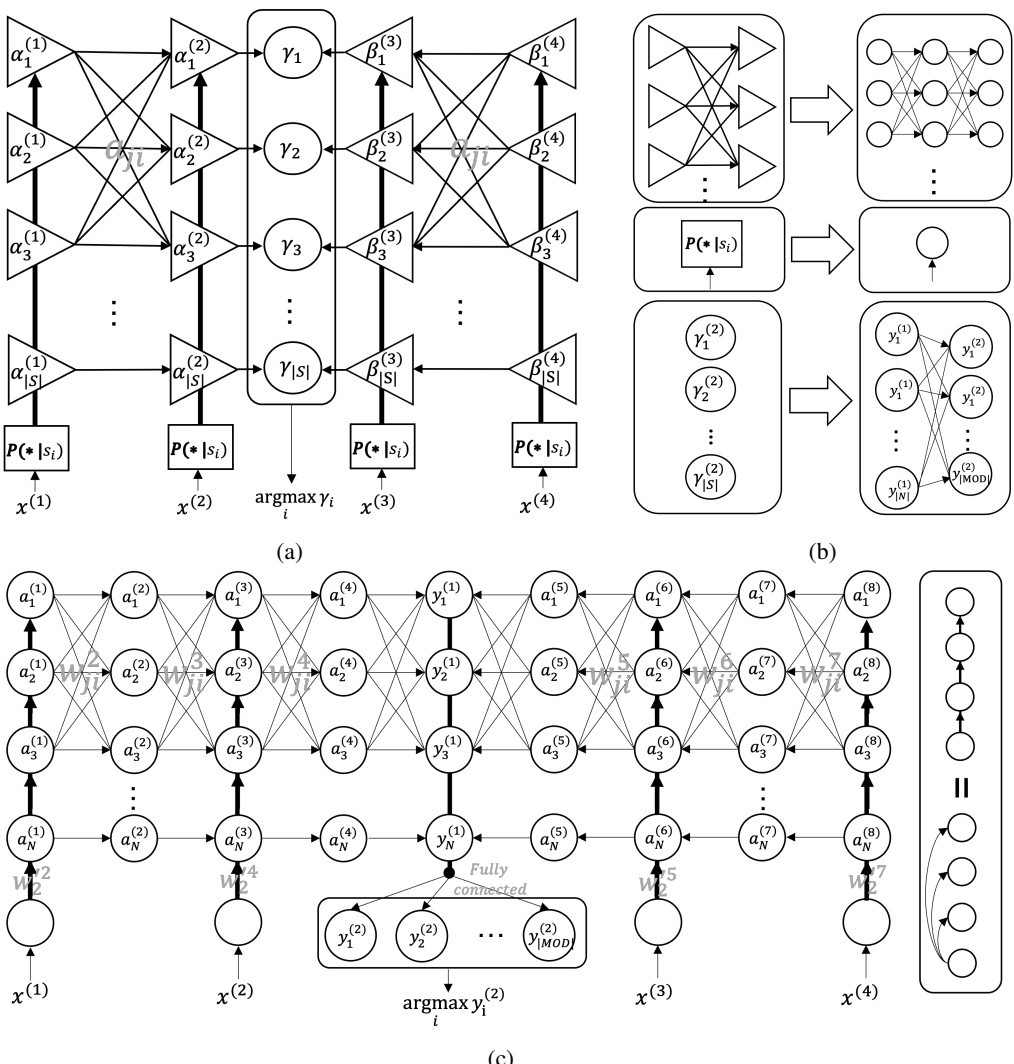

Figure 2: (a) Computational graph of FB algorithm, (b) Mapping FB to neural network: i) two consecutive products is replaced with two fully connected layers, ii) computing probability is replaced with a single perceptron, iii) two consecutive products to compute gamma is replaced with two fully connected layers, (c) Proposed neural network based equalizer, NeuralEQ. (Note that thick lines are used to represent full connections between components.)

2019). The fully connected structure has the advantage of being able to mimic any function, but, as can be inferred from the name, it often has unnecessarily many connections to implement the target function. There is another way to utilize RNN. Many recent works such as (Ye et al., 2018; Zhou et al., 2019; Kechriotis et al., 1994; Kim et al., 2020; Gomez Diaz et al., 2022) implement equalizer as a neural network, RNN is often used because RNN has excellent performance as a sequence detector, and equalizer can be seen as a kind of sequence detector. In fact, although previous studies have implemented equalizers that perform well through RNN, RNN has the following two problems. First, because RNN has a recurrent structure that requires past results in the current operation, it is difficult to satisfy timing margin when designing hardware requiring a high-speed operation, as in wireline communication. Second, because of the recursive structure, the same weight is repeatedly multiplied by the output, which causes gradient vanishing or expanding problems, making training difficult (Pascanu et al., 2013). Therefore, in view of these drawbacks, we develop a novel neural network architecture, neither the fully connected structure nor RNN, and the proposed architecture is inspired by the FB algorithm.

### 3.1 STRUCTURE DESCRIPTION

The proposed neural network is designed to have a similar structure to the FB algorithm. Figure 2a shows an illustration of the FB algorithm computation assuming that one symbol is decoded from four received values. As expressed in Equation 2, $\alpha^{t+1}$ is composed of the multiplication and sum of $\alpha_i^t$, $a_{ji}$, and $P(x|s_i)$, which are expressed by a line connected to the triangle in Figure 2a. $\beta^t$ calculated in the backward path is also drawn in the same way, and only the direction is opposite. Finally, the part where $\alpha$ and $\beta$ are multiplied to obtain $\gamma$ and find the state with the highest value is shown in the middle of Figure 2a.

As shown in Figure 2a, when recovering the target symbol given a sequence of channel output, values further away from the target need to go through more complex calculations compared to those closer. However, in case of fully connected neural nets, they have uniform depth for all inputs. Thus if they are used to emulate the FB algorithm, parameters would be wasted for the inputs close to the target symbol. Therefore the proposed neural network architecture reflects this imbalance in computing depths.

To replace each component of the Figure 2a with a neural network, we first note that, in the part $\alpha_i^t = \sum_{j \in |\mathbb{S}|} \alpha_j^{t-1} a_{ji} P(x^t|S^t = s_i)$, $\alpha_j$, $a_{ji}$, and $P(x|s_i)$ are multiplied in serial. $P(x|s_i)$ and $\alpha_j$ are functions of observed sequence $x$, and $a_{ji}$ is a constant given the system. We found that two fully connected layers are sufficient to learn this function. Then, the parts calculating $P(x|s_i)$ is replaced with a single perceptron. Finally, we replace the part Equation 1 with two fully connected layers whose output layer has the size of the number of modulation level, $|MOD|$. For example, if PAM4 is used, then the output layer has four neurons. These replacements are described in Figure 2b. Note that the number of neurons per layer, which is expressed as $N$ in Figure 2c, is not the same as $|\mathbb{S}|$ in the proposed architecture. Smaller $N$ is desirable to reduce the amount of computation, which will be described in more detail in Section 3.3

The main problem of implementing RNN-based equalizers in high-speed wireline communication systems is that the recurrent structure incurs huge area and power overhead. On the other hand, the proposed neural network does not have any feedback structure, so efficient pipelining in implementation is possible. DNNs composed of only fully connected layers also have the same benefits, but they are overparmeterized, causing a large computational and power overhead. Since the proposed architecture contains much less parameters to obtain the same performance as those with fully connected layers, it can be implemented with less complexity.

### 3.2 CONSIDERATIONS ON INPUT SIZE AND TARGET SYMBOL POSITION

In principle, the FB algorithm can decode the symbol at any location from input sequence. Even though Figure 2a describes the case when decoding the second symbol, it can calculate $\gamma$'s for any position between 1 and 4. However, it will not perform equally for all positions because the channel ISI can have both pre-cursors and post-cursors around the main-cursor, and for a given length of input sequence, as the target symbol position moves to the left (or right), the information on the pre-cursors (or post-cursors) disappears. In other words, position of the target symbol must be carefully chosen for good BER performance. In this study, we use the length of the input sequence and the target symbol position as 12 and 4, respectively. This generally showed good performance over various wireline channels tested. For the channels with much larger/less ISI compared to the tested ones, the input sequence length and target symbol position should be adjusted.

Let the channel input sequence be a vector $\mathbf{z}$, the signal with noise added to the channel output be vector $\mathbf{x}$, and the channel impulse response be $\mathbf{h}$, that is, $\mathbf{x} = \mathbf{z} * \mathbf{h} + \mathbf{n}$, where $\mathbf{n} \sim \mathcal{N}(0, \sigma^2 \mathbf{I})$. Also, the length of the input sequence is defined as T and the target symbol position is defined as $D$, the estimator $\hat{\mathbf{z}}$ for the original signal $\mathbf{z}$ can be expressed as $\hat{z}^{t+D} = f_{\mathbf{w}}(x^{t:t+T-1})$. Note that $f_{\mathbf{w}}$ indicates the function of proposed neural network, NeuralEQ. However, if there are pre-cursors in the channel ISI, delay occurs between $\hat{\mathbf{z}}$ and $\hat{\mathbf{x}}$, the final equation is summarized again as follows.

$$\hat{z}^{t+D-|pre\_cursors|} = f_{\mathbf{w}}(x^{t:t+T-1}) \tag{4}$$

where $|pre\_cursors|$ is the number of pre-cursors in ISI.

## 3.3 COMPLEXITY

For the proposed neural network to be legitimate, the proposed NeuralEQ should be less complex than the original FB algorithm. To analyze complexity, we only compare the forward path equation of the FB algorithm and NeuralEQ. Even though both the FB algorithm and NeuralEQ consist of forward path, backward path, and gamma computation, forward and backward paths have identical structures and the computational load of gamma is ignorable. Analyzing results are summarized as Table.1.

The computation type and number of operations required to calculate $\alpha_i^t$ from $\alpha_j^{t-1}(j \in \mathbb{S})$, and $a_i^{t+1}$ from $a_j^{t-1}(j \in N)$ are shown in Table.1. As shown in Figure 2b, because one layer of the FB algorithm is replaced with two layers of a neural network, the forward-path equation of NeuralEQ is also a two-layer computation. i.e. $a^{t-1}$ to $a^{t+1}$. The number of multipliers of the FB algorithm, which is a dominant factor in computation complexity, is $|\mathbb{S}|^2 + |\mathbb{S}|$. On the other hand, NeuralEQ has $2N^2 + N$ multipliers.

If N and S are the same, NeuralEQ has more multipliers, but it is not. $|\mathbb{S}|$ is determined by $|MOD|^{|\mathbf{ISI}|}$, where the length of ISI is 10 and the case of PAM4 is $4^{10} = 2^{20}$. However, NeuralEQ has a good performance for the length of ISI equals 10 even when $N = 32$. In this case, the number of multipliers is $2^{11} + 32$, which means NeuralEQ has $2^9$ times fewer multipliers. In addition, we will discuss a method to further reduce complexity in Section.4.2, which shows further compression of the NeuralEQ's complexity.

| Feature | FB algorithm | | NeuralEQ | |
|---|---|---|---|---|
| forward-path equation | $\alpha_i^t = \sum_{j \in |\mathbb{S}|} \alpha_j^{t-1} a_{ji} P(x^t|s_i)$ | | $a_i^t = \tanh(\sum_{j \in N} a_j^{t-1} w_{ji}{}^t$ $+ M w_{N+1,i} + b_i^t)$ , where $M = \tanh(w'^t_i x^t + b'^t_i)w_i^t)$ $a_i^{t+1} = \tanh(\sum_{j \in N} a_j^t w_{ji}^{t+1} + b_i^{t+1})$ | |
| # of multipliers | $|\mathbb{S}|^2 + |\mathbb{S}|$ | $1.10 \times 10^{12}$ | $2N^2 + N$ | $2.08 \times 10^3$ |
| # of adders | $|\mathbb{S}|^2 - |\mathbb{S}|$ | $1.10 \times 10^{12}$ | $2N^2 + N$ | $2.08 \times 10^3$ |
| # of conditional prob. | $|\mathbb{S}|^2$ | $1.10 \times 10^{12}$ | N/A | N/A |
| # of tanhs | N/A | N/A | $3N$ | $9.6 \times 10^1$ |

Table 1: Comparison of the number of computations between FB and NeuralEQ. The forward-path equation of NeuralEQ is derived for time:t+1 from time:t-1, unlike the FB algorithm. It is because one layer of the FB algorithm corresponds to two layers of NeuralEQ as shown in Figure 2. Also, notations are referred by Figure 2. Then the numerical values were calculated for the case where PAM4 modulation is used, the length of ISI is 10 and N = 32.

## 3.4 TRAINING AND PERFORMANCE

The loss function of the proposed neural network is defined as follows, using the notations in Equation.4.

$$loss = \frac{1}{N} \sum_{t=1}^{N} crossEntropy(f_{\mathbf{w}}(x^{t:t+T-1}), z^{t+D-|pre\_cursors|}) \tag{5}$$

where N is the batchsize. In general, the trained network may not be generalized well due to the limited number of the training data. Even the proposed neural network shows low performance, if not provided various data patterns, the correlated ISI, and the various noise environments as examples. Therefore, securing a large number of training sets $x^{t:t+T-1}$ and $z^{t+D-|pre\_cursors|}$ is advantageous for training. The good news is that in wireline communication, as long as data is continuously transmitted from the transmitter, it is possible to generate x and z with infinite noise patterns and various ISI patterns. Because data could be infinitely generated, there was no reason to repeatedly use the same training data. Thus we have no notion of 'epoch', or rather, the network was trained in a single epoch. Every time training data was needed, it was freshly generated. Training, validation and evaluation sets were each randomly generated with the same ISI and SNR as parameters. The only difference is their size. Sequences of length 2e9, 2e8, 1e7 were used for training, validation, and

evaluation respectively. Every time 1e7 training data was used, we used 1e6 of validation data for validation. The training parameters are summarized in Table.2.

To examine the performance of the proposed NeuralEQ, Figure 3 can be obtained with the same environment as set in Figure 1b, i.e., ISI=[1.0,0.4,0.2,0.1], the number of FFE taps is 8 and the number of DFE taps is 3. It is shown that the neural network has superior performance to FFE and DFE, and has performance close to optimal decoder, the FB algorithm. In addition, results for various channel environments are confirmed in Section.4 , and all of them have better performance than conventional EQs.

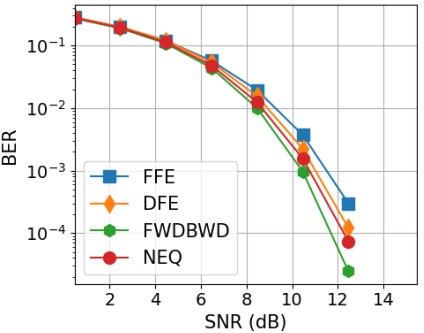

Figure 3: Performance of NeuralEQ

| Network parameters | Value |
|---|---|
| Input size | 12 |
| Target symbol position(D) | 4 |
| # of neurons per layer(N) | 32 |
| **Training parameters** | **Value** |
| # of train, valid, eval sets | 2e9, 2e8, 1e7 |
| Batchsize | 8192 |
| Loss function | Cross Entropy |
| Optimizer | Adam |
| Learning Rate | 1e-3 |

Table 2: Proposed NeuralEQ hyper-parameters

## 4 EXPERIMENT RESULTS

### 4.1 PERFORMANCE ON VARIOUS CHANNEL

Four different channels were prepared to evaluate the performance of the proposed NeuralEQ. Each channel had -7dB, -12dB, -16dB, and -21dB loss in Nyquist, respectively, and SBR was extracted from each channel and used for simulation. (refer to Appendix A) In addition to NeuralEQ, the combination of FFE and DFE, which are conventional EQ, was also evaluated. Unlike the channels used in the section.3, channels in this section have pre-cursors, which is more realistic, so it is challenging to perform properly by FFE or DFE alone, so they are used together and compared with NeuralEQ. In order to perform evaluation, the number of FFE taps is set to 24, and the number of DFE taps is 5, which is more than what is actually used in the PAM4 wireline application so that there is no performance degradation due to lack of taps. The parameters of NeuralEQ are the same as those shown in Table.2, except for the ISI of the -21 dB channel, which has worse ISI than other channels, so in this case, the length of the input sequence is increased from 12 to 24.

As illustrated in Figure 4, it can be shown that the performance of the proposed NeuralEQ in the whole SNR region is superior to the combination of FFE and DFE with sufficient taps. In particular, the gap in performance increases as the ISI, which show that NeuralEQ is more valuable than existing EQ in recent wireline communication that suffers more from high channel loss.

### 4.2 PRUNING

Although the proposed NeuralEQ needs multiple symbols per a single decoding, not all input symbols have the same information in decoding the target symbol. This is because the magnitude of the cursors decreases as the distance away from the main-cursor, so the information of the target symbol is minor. On the other hand, the symbol at the main-cursor position has the most information. Due to this imbalance in information of symbol position, even if the complexity of the left and right side layers of the NeuralEQ is lower than those of the middle, it can be expected that NeuralEQ will not significantly degrade performance.

The pruning (Janowsky, 1989; Karnin, 1990) is the efficient methodology to strike a balance between the difference of information and computational complexity, reducing unnecessary neurons. Pruning has been studied in a way that reduces hardware complexity without performance degradation from a model that has been trained. In particular, (Frankle & Carbin, 2019) consists that there exist a

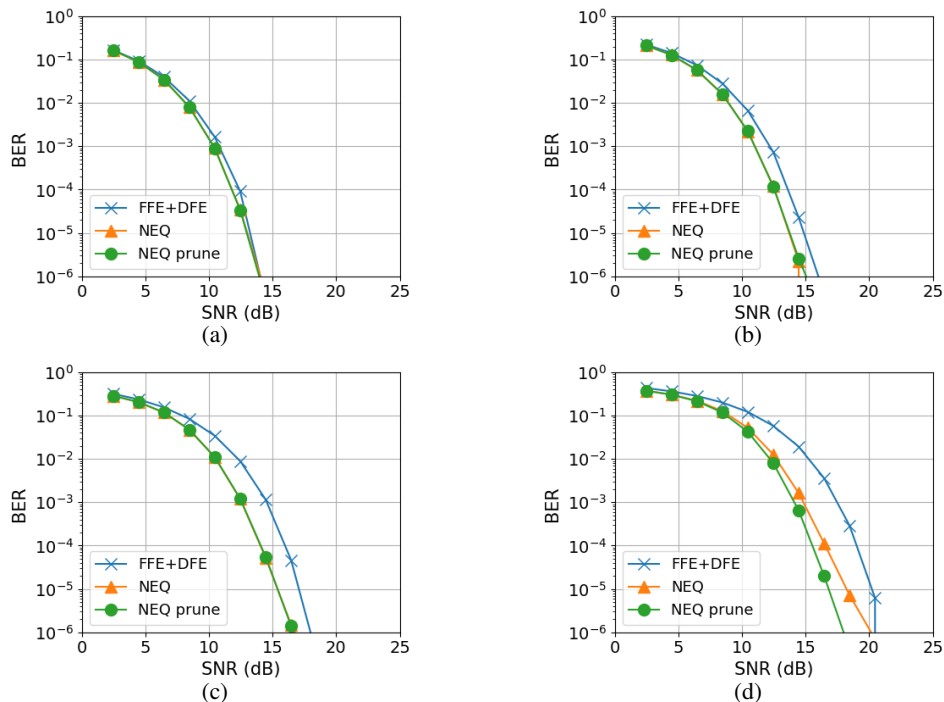

Figure 4: Performance comparison of FFE+DFE, NeuralEQ, and NeuralEQ with pruning, for channel loss (a) -7dB, (b) -12dB, (c), -16dB, and (d), -21dB. In all cases, NeuralEQ has better performance than FFE+DFE with or without pruning. Especially for channel loss -21dB, pruning increases the performance of NeuralEQ.

model which has better or similar performance even after pruning, so we looked at whether pruning can improve its performance even for the proposed NeuralEQ.

Figure 5 shows how much the performance degrades as the progress of pruning. Every pruning iteration, 10% of the total weight which are the smallest ones is pruned. The Y-axis is the normalized value of BER before pruning, and the x-axis represents sparsity. Performance degradation due to pruning appears differently for each channel. When the ISI is small, for -7dB lossy channel, performance degradation due to pruning is small, and when the ISI is large, it is relatively sensitive to pruning. Also, Appendix C describe how the sparisty of each layer varies with the progress of pruning.

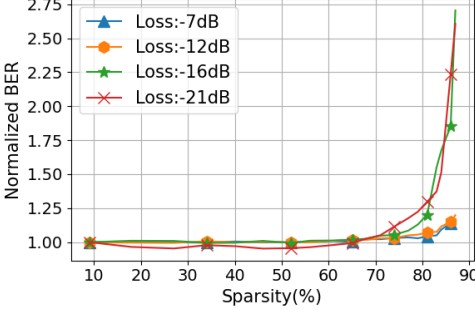

Figure 5: Normlized BER via sparsity

| Channel loss | # of param. before prune | # of param. after prune | Diff. |
|---|---|---|---|
| -7dB | 38,564 | 13,446 | -65% |
| -12dB | 38,564 | 16,600 | -57% |
| -16dB | 38,564 | 18,445 | -52% |
| -21dB | 76,964 | 36,812 | -52% |

Table 3: NeuralEQ parameter reduction after pruning for various channel

In Figure 4, the performance is reevaluated using the pruned model. It can be seen that there is no significant difference between the performance before pruning or that it has better performance for the -21 dB channel. The number of parameters before and after Pruning is summarized in

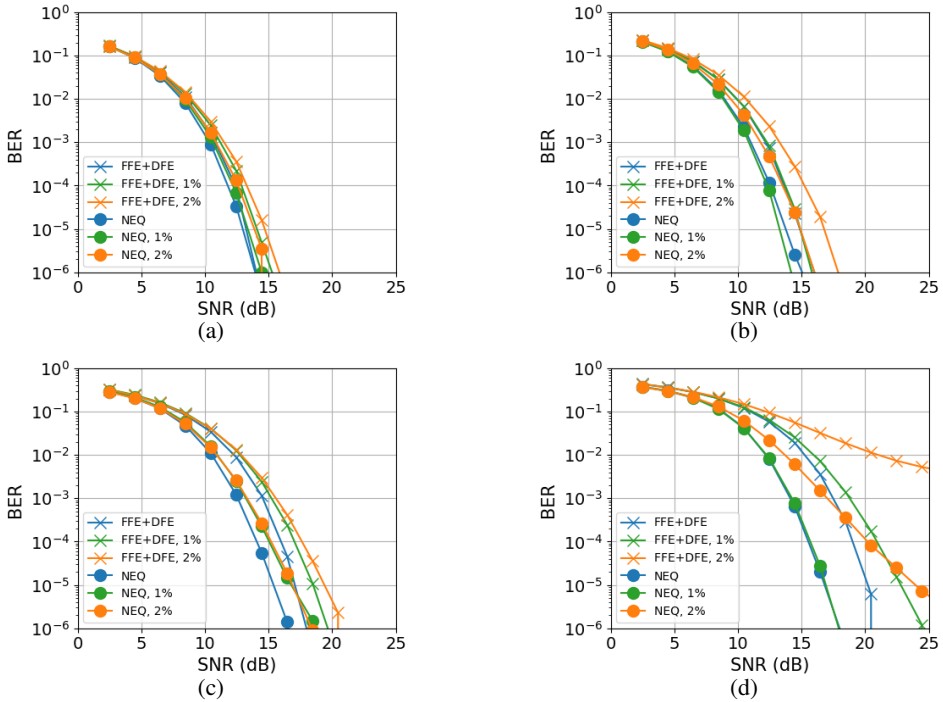

Figure 6: Performance variation due to ISI skew for channel loss (a) -7dB, (b) -12dB, (c) -16dB, and (d) -21dB, which occurs in practical application. Both conventional EQ and NeuralEQ are evaluated for comparison, and the magnitude of skew of ISI is varied with 0%, 1%, and 2%. Regardless of ISI skew, NeuralEQ has better performance than conventional EQ.

Table.3. Accordingly, pruning is a well-suited methodology to reduce the number of parameters needed adaptively for each channel without degradation.

### 4.3 ROBUSTNESS TO ISI VARIATION

In practice, the channel environment during training and after training could be different. Therefore, test sets which has skew on ISI during training are generated and then we evaluate the performance degradation. The combination of FFE and DFE is also evaluated in the same environment as a control group. The amount of skew to ISI is as follows. When the single bit response of the channel is $\mathbf{h}$ and let skew be $\mathbf{s}$, $\mathbf{h_{skew}} = \mathbf{h} + \mathbf{s}$, where $\mathbf{s} \sim \mathcal{N}(0, \sigma^2 \boldsymbol{I})$, and $\sigma = \max(\mathbf{h})p$ $(p \in \{0, 0.01, 0.02\})$, which is proportional to the magnitude of main-cursor.

Figure 6 shows how much the performance of conventional EQ and NeuralEQ degrades when the skew is generated to ISI for each channel. In the case of 7 dB, skew does not affect performance much, but it can be seen that the degree of degradation increases as channel loss increases. It may not be deemed that NeuralEQ is stronger against ISI skew than conventional EQ, but it is not more sensitive. The performance superiority of traditional EQ and NeuralEQ is not reversed despite skew, and it can be confirmed that NeuralEQ has acceptable tolerance to ISI variation.

## 5 CONCLUSION

In this paper, we developed a neural-network-based equalizer suitable for high-speed wireline communication. It was shown that the proposed neural EQ mimics the FB algorithm, reduces computational complexity, and has superior performance to the existing equalizer. In addition, since there is no feedback loop, the high-speed design is easier than the existing equalizer, such as DFE. The pruning technique was applied to further reduce complexity without performance degradation for more efficient hardware implementation. Also, we verified robustness to ISI variation is acceptable compared to the existing EQ.

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

## A   CHANNELS FOR SIMULATION

The channels for performance evaluation of the proposed NeuralEQ are extracted from real PCB striplines with different lengths. Channels of loss 7dB, 12dB, 16dB, 21dB are from PCB stripline of length 3inch, 5inch, 7inch, 9inch respectively. The characteristics of the channels are shown in Figure7. Each channel has a loss between 7 dB and 21 dB at nyquist frequency(Figure.7a), and the SBR of each channel is shown in Figure.7b. The SBR of the channel extracted values only if it is greater than 0.001 times of the main-cursor, otherwise, it is forced to zero. The higher loss channel, the longer ISI it has.

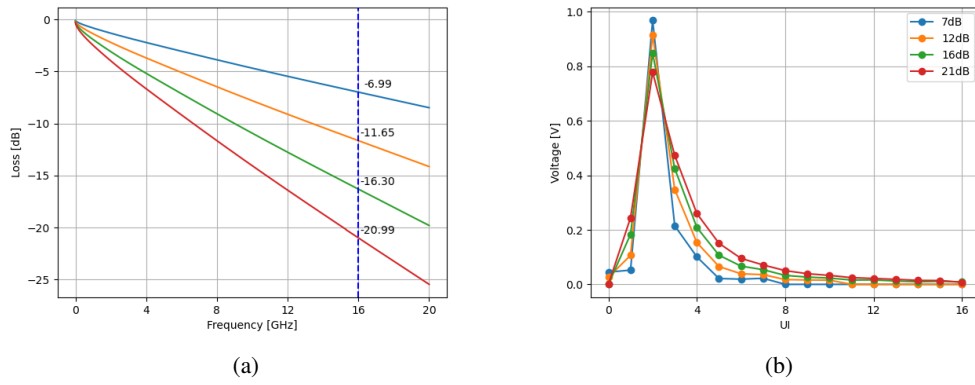

(a)                                         (b)

Figure 7: (a) Frequency response, and (b) single bit response of tested channels

## B   DECISION OF SNR FOR TRAINING

When creating a data set for training, we must consider the amount of noise applied to the data. If noise is not applied at all, the neural network has not learned about the input sequence with noise, so noise will not be effectively removed during decoding. If the noise of the training set is too large, it is highly likely that the network will not converge to the optimal point.

Figure.8 shows how the performance changes by changing the noise applied during training, that is, SNR. As expected, too large or too small noise degraded performance, and when trained for SNR with a BER of about 1e-2 empirically, a neural network with good overall performance was obtained.

It should be noted that the noise applied to the test set and the noise applied to the training set does not produce a good performance. That is, the SNR used during training to have good performance does not match the SNR to be applied during the test. For example, the performance at 18dB SNR is better when trained with 12dB SNR than the result of training at 18dB SNR in Figure.8d. Simulation results for various channels show that training SNR having performance around 1e-2 BER is used in a training phase, the network is learned to have better performance in a full range of SNR, and in this paper, these SNRs were found and trained for each channel.

## C   SPARSITY OF EACH LAYER WITH PRUNING PROGRESS

When pruning is performed on NeuralEQ, having small weight, low importance neurons, will be removed and high importance neurons will remain. Since ISI usually has a smaller cursor value as it goes toward the edge around the main-cursor, it can be assumed that the layers at the edges are less important than the middle even in NeuralEQ.

Figure.9 shows the NeuralEQ pruning for -7dB, -12dB, -16dB, and -21dB channels and the sparsity of each layer for each pruning. The difference in sparsity between layers means the information

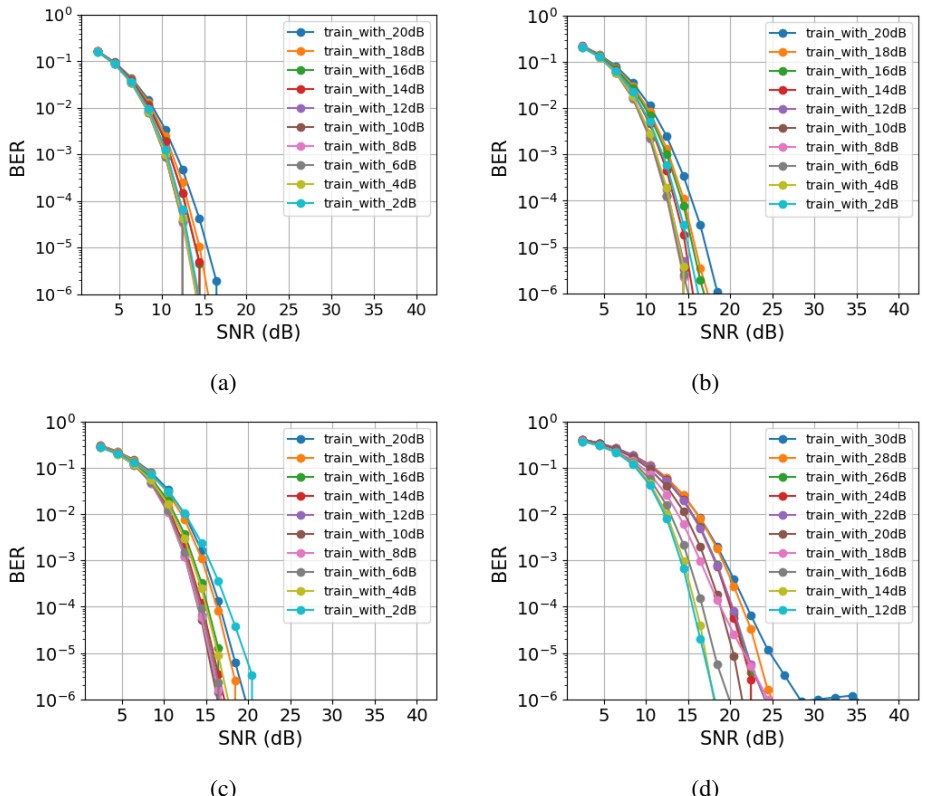

Figure 8: Performance comparison via varying training SNR. Various channels are evaluated with (a) -7dB, (b) -12dB, (c), -16dB, and (d), -21dB loss.

per symbol is not important with the same amount for decoding. Pruning progress can remove unimportant neurons efficiently.

## D ROBUSTNESS TO SIGNAL QUANTIZATION

The quantization of channel signals must be preceded to implement the proposed NeuralEQ into digital hardware. For such implementation to be feasible, it must be possible to limit the performance degradation due to quantized inputs, while using a reasonable number of bits for quantization. We anticipated the degradation, and the appropriate number of bits, by training and evaluating the NeuralEQ with quantized channel signals.

The performance of NeuralEQ with quantized channel signals, for channels of loss -7dB, -12dB, -16dB, and -21dB, are shown in Figure.10. The quantization range was set as the range of the channel signal without probabilistic noise, given the channel loss. The quantization bit number was set between 5 to 8 bits. Other training and evaluation parameters were unchanged. The performance of pruned NeuralEQ without quantization is also shown for comparison.

## E COMPARING COMPLEXITY OF NEURALEQ WITH FC NEURAL NETWORK

In this section, we evaluate how the proposed NeuralEQ efficiently mimics the optimum decoder, comparing it with the fully connected network. The fully connected neural network can also mimic an arbitrary function, but the problem is that the complexity is too high. By quantifying the difference in complexity to achieve target performance, we can claim the efficiency of the proposed NeuralEQ.

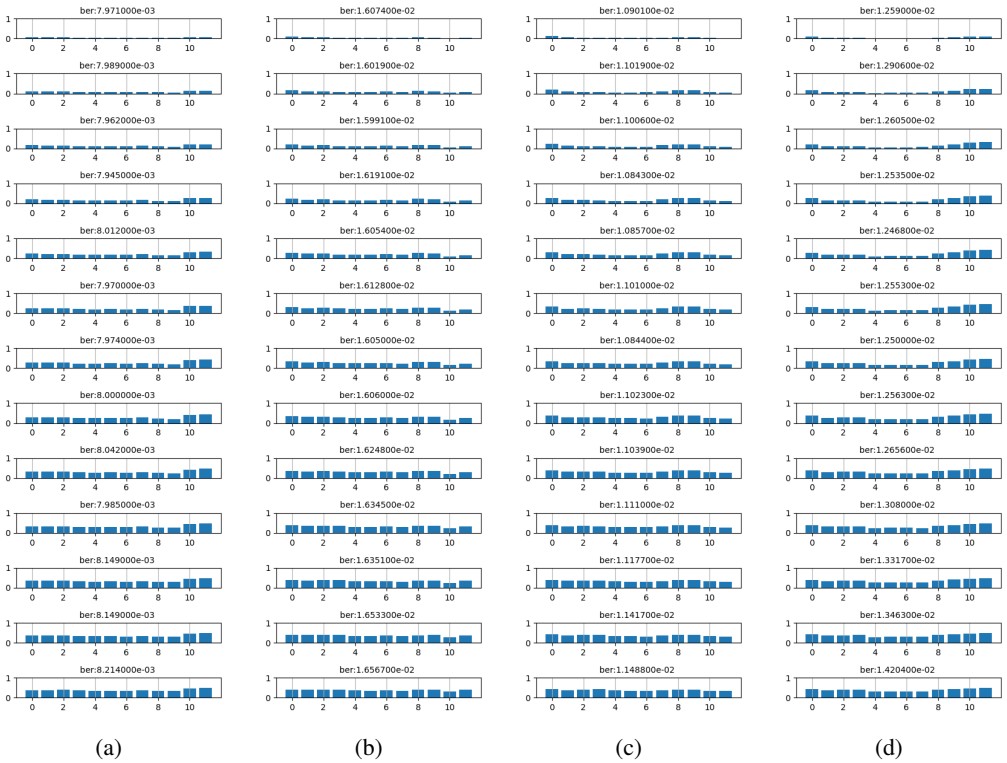

Figure 9: Sparsity of each NeuralEQ layer with thirteen pruning iterations. (a) Pruning for -7dB channel, (b) -12dB channel, (c) -16dB channel, (d) -21dB channel. 10% of total weight is removed per iteration and corresponding BER is written for each iteration.

The proposed NeuralEQ has a hyperparameter $N$, which is the number of features per layer. Choosing an optimum $N$ is indeed an important process to go through. There are many ongoing studies about optimizing hyperparameter of the network and researchers have provided different types of search strategies depending on network structure. We choose Tree-Parzan Estimator (TPE) because it is appropriate algorithm for model which has low dimensional search space and NeuralEQ has a single hyperparmeter $N$.

As a result, Table.4 shows that the proposed NeuralEQ outperforms FC network for all four channels conditions. NeuralEQ not only provides better BER than FC network does, but also implements fewer parameters than FC network does to run the network.

| SBR (dB) | FC Network | | | NeuralEQ | | |
| | Structure | BER (e-3) | Parameters | Structure | BER (e-3) | Parameters |
| --- | --- | --- | --- | --- | --- | --- |
| 7 | [ 216, 376 ] | 1.126 | 85,908 | N: 22 | 0.961 | 18,594 (21.6%) |
| 12 | [ 408, 488 ] | 2.845 | 206,852 | N: 20 | 2.309 | 15,464 (7.47%) |
| 16 | [ 472, 344 ] | 12.56 | 170,228 | N: 35 | 11.221 | 45,959 (27.0%) |
| 21 | [ 352, 512 ] | 56.054 | 187,364 | N: 29 | 50.516 | 31,817 (16.9%) |

Table 4: TPE Strategy Results

# F    SIMULATION RESULTS FOR ADDITIONAL CHANNELS

In addition to the channels used earlier, manually modified channels are used to further evaluate the performance of the proposed NeuralEQ. The more diverse channels are shown in Figure 11a. Compared to Figure 7b, the number of pre-cursors is increased, and the polarity of random cursors

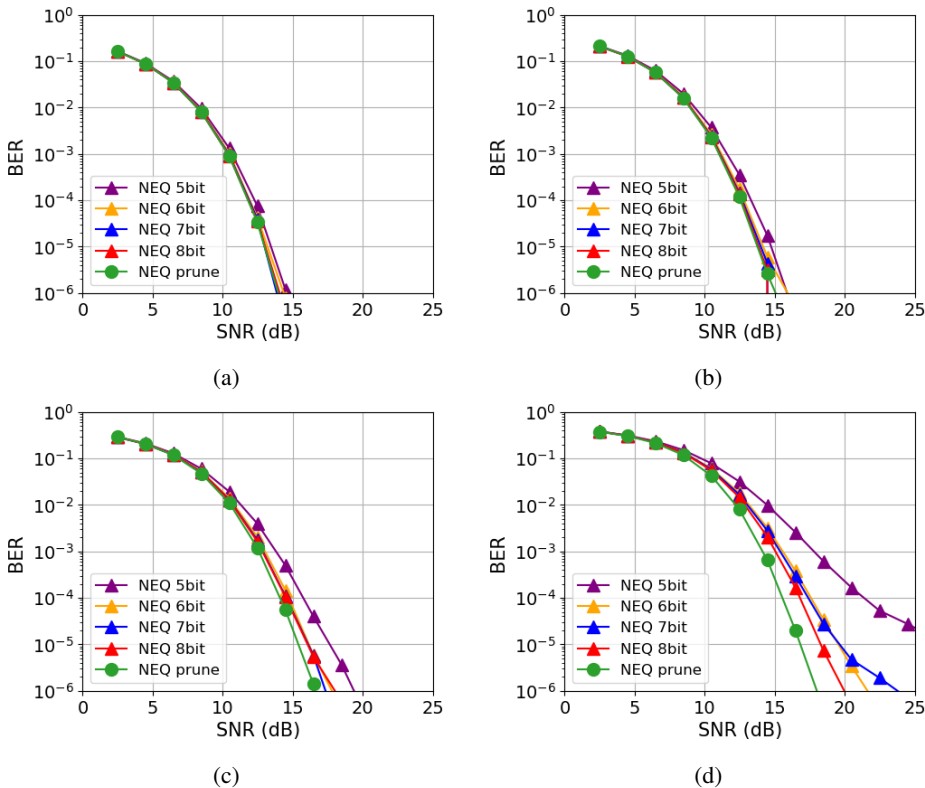

Figure 10: Performance for different quantization bit numbers, for channels of loss (a) -7dB, (b) -12dB, (c) -16dB, and (d) -21dB. The performance of pruned NeuralEQ without quantization is added for comparison.

are flipped. The length of ISI is large in the order of Custom_CH3, Custom_CH2, Custom_CH1. The simulation performed in Figure 4 is performed again for the new channels. The results are shown in Figure 11. It is confirmed that the proposed NeuralEQ has better BER performance than conventional EQ for the newly generated channels too.

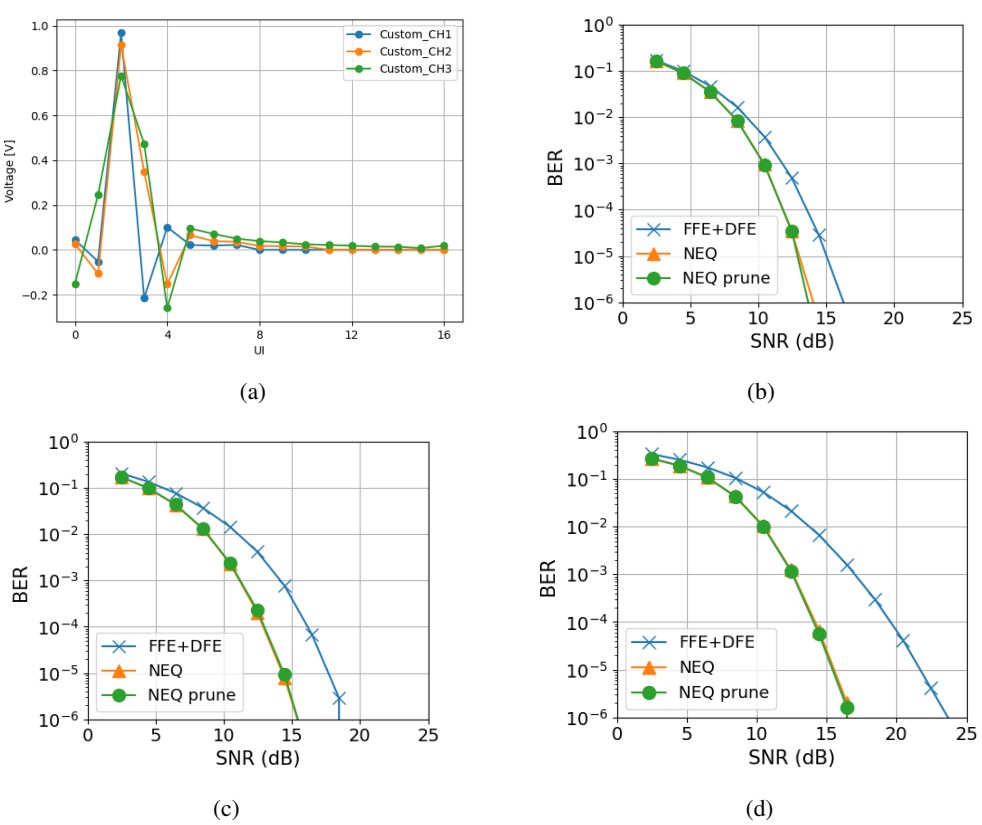

(a)

(b)

(c)

(d)

Figure 11: Performance for customized channel which has more pre-cursor and negative ISI. (a) Single bit response of each customized channel. BER performance for (b) Custom Ch1, (c) Custom Ch2, and (d) Custom Ch3.

