# OpenReview forum: "NeuralEQ: Neural-Network-Based Equalizer for High-Speed Wireline Communication"
_ICLR.cc/2023/Conference — Submitted to ICLR 2023_

### Official Review · Reviewer_Qiy5 · 2022-10-22

**Confidence:** 3
**Correctness:** 3
**Technical Novelty And Significance:** 3
**Empirical Novelty And Significance:** 2
**Recommendation:** 6

**Clarity, Quality, Novelty And Reproducibility:**

These are good in this paper.


**Strength And Weaknesses:**

Strengths:

—-------------------

The paper has many strengths and is tackling an important problem. For the sake of time  I am only writing about weaknesses in this review, since those are the ones that should be actioned upon.

Weaknesses (W)

—--------------------

W: this paper is about applying neural network to replace a specific algorithm, namely the equalizer of a wired connection receiver. This makes the impact of the paper low for the machine learning community. Which of the finding can be generalized to other domains and how? If this cannot be described, maybe this paper is better suited in the publications in the communications domain.

W: The authors do not extensively study the reasons behind the good performance of the NeuralEQ nor they discuss how the resulting NeuralEQ is different from the Forward-Backward algorithm, from which it is motivated. It is clearly different since it has lower complexity in many cases. Is it because of some assumptions in the Forward-Backward algorithm are wrong?

W: The authors do not discuss the training and validation data split in detail. The training and validation data should be described in detail in every setting.

W: How general is the trained algorithm? Does the algorithm need to be retrained if something in the transmission path changes? What are the typical changes? How often in real life would this happen?

W: How representative are the simulations to the real world? Since all results are produced by simulations, the authors should cite previous works that have compared similar simulations to real channels. The best option would of course be to record real data and validate the simulation trained algorithm on real data.


Minor weaknesses and comments:

—---------------------

Abstract: “Rapid growth of ML applications demands high-performance computing systems
to perform massive data processing.”, I don’t think ML applications are the only ones that require high bandwith, e.g., video applications is pretty large.

Minor spelling mistakes:

Header 4: “EXPERMIENT RESULTS” -> EXPERIMENT RESULTS

4.2 “NerualEQ” -> NeuralEQ

Figure 4: “pruning increase
the performance of NeuralEQ.” 🡪 … pruning increases the performance of NeuralEQ


**Summary Of The Paper:**

UPDATE: I have read the rebuttal and updated the score.

XXXXXXXXXXXXXXXXXXX


This paper addresses the problem of extremely fast wired connections, such as in very fast ethernet or PCIe connections. The main issue when increasing the communication speed is ISI (inter-symbol interference), which is traditionally solved using an equalizer. A very good equalizer is the forward-backward algorithm, but it is prohibitively high in computational complexity. This paper presents a learned equalizer, which has lower computational complexity in some cases, while retaining good bit error rate performance.


**Summary Of The Review:**

Relatively good work with some flaws that would warrant a new revision.

---

> ### Author Response · Authors · 2022-11-15
> **Respond to Reviewer Qiy5 (1/2)**
>
> __W1__ This paper is about applying neural network to replace a specific algorithm, namely the equalizer of a wired connection receiver. This makes the impact of the paper low for the machine learning community. Which of the finding can be generalized to other domains and how? If this cannot be described, maybe this paper is better suited in the publications in the communications domain.
>
> __Answer to W1__ Thank you for your detailed comments. Our paper is indeed focused on a certain application of machine learning: utilizing neural networks to enhance the equalization performance of wireline communication. To our knowledge, ICLR also calls for papers on the applications of ML, and many on such topics have been accepted in the past. We believe that the category suits our paper well.
>
> Moreover, our paper doesn't simply apply the well-known neural network structures(e.g., FC, RNN, transformer, etc.) to solve a problem. We analyzed the computations needed for the target problem, created a neural net structure that can learn them efficiently, and showed that it performs better.
>
> What if we had used a FC network? As described in Section 3.1, with FC networks, parameters would be wasted for inputs close to the target symbol. This not only increases the hardware complexity, but also makes it more difficult to find the optimal parameters. This can be seen in Appendix E, where the BER performance and complexity of a FC network and NeuralEQ are compared. NeuralEQ shows better BER performance with much smaller complexity.
>
> Instead of applying conventional structures to solve a problem, we started with a classic mathematical method (the FB algorithm, in this case). Then we devised a neural network structure with lower complexity but with similar computation depths (the NeuralEQ). We believe that such an approach can be practiced in other ML applications and that it can show value at this conference.
>
>
> __W2__ The authors do not extensively study the reasons behind the good performance of the NeuralEQ nor they discuss how the resulting NeuralEQ is different from the Forward-Backward algorithm, from which it is motivated. It is clearly different since it has lower complexity in many cases. Is it because of some assumptions in the Forward-Backward algorithm are wrong?
>
> __Answer to W2__ The reason why NeuralEQ has good performance is that the NeuralEQ architecture is inspired by the structure (or computational graph) of the FB algorithm, which we believe allows NeuralEQ to mimic the behavior of the FB algorithm by training. As a result, despite the low complexity of NeuralEQ, it is able to achieve the performance close to the FB algorithm. Note that, since the FB algorithm computes the posterior probabilities with given observations and selects the symbol with highest probability, FB has the optimal BER performance for symbol detection. However, the huge computational complexity of FB prevents it from being used in wireline communication.
>
> Motivation of our study was to find an equalizer with much lower complexity than FB but still with better BER performance than conventional EQ (FFE, DFE). Since conventional EQ and FB have their own limitations (noise boosting in FFE, error propagation in DFE, complexity in FB), we expect that a new solution could be found using neural networks. While conventional EQ and FB have either extremely low complexity/high BER or extremely high complexity/low BER, the proposed NeuralEQ bridges the gap, allowing us to obtain a simpler equalizer than the FB algorithm, while having better BER performance than conventional EQ.
>
> For the comparison with the FB algorithm, a BER curve can be seen in Figure 3, which is a simulation result for a simple channel model. However, the experimental results for real channels (Figure4 and Figure6) do not show the performance of the FB algorithm as you claimed. This is due to the complexity of the FB algorithm. The number of states of the FB algorithm is exponentially proportional to the length of ISI, and the real channels used in the experiment have more than 10 post-cursors, which requires too large memory of the simulation machine. In contrast, the channel used in Figure3 has only 3 post-cursors. The real channels required 4^7 times more memory at least than the channel in Figure 3. It implies that even if the FB algorithm has the optimal performance, it is not suitable for applications with a very large number of states, such as ultra high-speed wireline communication. It also emphasizes how effective the proposed NeuralEQ is in such applications.

---

> > ### Author Response · Authors · 2022-11-15
> > **Respond to Reviewer Qiy (2/2)**
> >
> > __W3__ The authors do not discuss the training and validation data split in detail. The training and validation data should be described in detail in every setting.
> >
> > __Answer to W3__ As described in Section 3.4, with a given ISI and SNR (channel type), it is possible to randomly generate data infinitely. The training, validation, and evaluation data have exactly the same properties in this manner. They were each randomly generated with the same ISI and SNR as parameters. The only difference is their size. Sequences of length 2e9, 2e8, 1e7 were used for training, validation, and evaluation, respectively. Every time 1e7 training data was used, we used 1e6 of validation data for validation.
> > This will be soon added to the paper.
> >
> > __W4__  How general is the trained algorithm? Does the algorithm need to be retrained if something in the transmission path changes? What are the typical changes? How often in real life would this happen?
> >
> > __Answer to W4__ If ISI of the channel (i.e., channel impulse response) changes, the algorithm needs to be retrained.
> > However, in real applications, the ISI of a wireline channel doesn’t change much but small variations (e.g. due to temperature and humidity change) may occur. We showed in the paper that our algorithm runs well for ISIs with such small variations(Figure 6).
> >
> >
> > __W5__ How representative are the simulations to the real world? Since all results are produced by simulations, the authors should cite previous works that have compared similar simulations to real channels. The best option would of course be to record real data and validate the simulation trained algorithm on real data.
> >
> > __Answer to W5__ Thank you for your comment. We realized we had not explained our simulation channels thoroughly in the paper. The channel models were actually created by using s-parameters extracted from real PCB striplines with different lengths. Thus the simulated channels are indeed based on real data.
> >
> >
> > __Answer to additional W__ Thank you for your kind notice! We fixed them accordingly and will upload the revised paper.

---

> > > ### Comment · Reviewer_Qiy5 · 2022-12-12
> > > **Thank you for the rebuttal**
> > >
> > > I have read the rebuttal and it has addressed many of the concerns by the reviewers and I have upgraded my score.

---

### Official Review · Reviewer_HhyM · 2022-10-23

**Confidence:** 3
**Clarity, Quality, Novelty And Reproducibility:** 1. It is an interesting idea to repla…
**Correctness:** 3
**Technical Novelty And Significance:** 3
**Empirical Novelty And Significance:** 2
**Recommendation:** 6

**Strength And Weaknesses:**

Strength:
The proposed solution has lower computational complexity than the forward-backward (FB) algorithm, while the BER performance is better than conventional equalizers.

Weakness:
1. The BER performance of the proposed solution is worse than the FB algorithm, especially in the high SNR region.
2. The authors did not provide a quantitative comparison between the proposed solution and the FB algorithm. The authors did not provide any detailed FLOPs numbers.

**Summary Of The Paper:**

This paper proposes a neural network-based equalizer that mimics the forward-backward algorithm for wireline communication. The authors also tried pruning the neural network.

**Summary Of The Review:**

Overall, the proposed idea is interesting. It has lower complexity than the optimal FB equalizer while achieving better BER performance than conventional equalizers. However, the authors did not provide a quantitative comparison for the computational complexity. How much more computation cost does the FB algorithm have compared to the proposed solution? Also, it is unclear whether the computation cost of the proposed solution is much higher than conventional equalizers.

---

> ### Author Response · Authors · 2022-11-14
> **Respond to Reviewer HhyM**
>
> __W1__ The BER performance of the proposed solution is worse than the FB algorithm, especially in the high SNR region.
>
> __Answer to W1__ Yes, we agree that the FB algorithm shows better BER performance than NeuralEQ when SNR is high. However, the FB algorithm is too complex to achieve the high data rates required in wireline communication. Thus, despite the higher BER, faster equalization methods such as FFE and DFE have been conventionally used.
>
> Such tradeoff between BER and throughput also applies to our method. As you mentioned, it has to sacrifice some BER. However, NeuralEQ has better BER performance compared to FFE and DFE(Figure 3, 4), while having a structure that can be efficiently pipelined, allowing it to provide very high data rates.
>
> We additionally showed that the computational complexity of NeuralEQ related to the size and cost of hardware is much lower than that of the FB algorithm(Table 1). So we acquired even more advantages from the BER tradeoff.
>
>
> __W2__ The authors did not provide a quantitative comparison between the proposed solution and the FB algorithm. The authors did not provide any detailed FLOPs numbers.
>
> __Answer to W2__ Thank you for pointing this out. We added the calculated numbers of operations to Table 1. When PAM4 is used and the length of ISI is 10, more than 10^8x operations are needed in the FB algorithm compared to NeuralEQ. Furthermore, the complexity of NeuralEQ can be decreased via pruning as shown in Section 4.2.

---

### Official Review · Reviewer_X5rc · 2022-10-24

**Confidence:** 4
**Correctness:** 2
**Technical Novelty And Significance:** 2
**Empirical Novelty And Significance:** 2
**Recommendation:** 3

**Clarity, Quality, Novelty And Reproducibility:**

A certain novelty can be recognized in the proposed approach, while the system and channel models are too much naive from a view point of implementation.

**Strength And Weaknesses:**

The proposed approach itself might have some novelty, however, I cannot recommend the acceptance
of the paper, mainly due to the lack of effectiveness.
Some detailed comments are listed below.

-From the view point of the development of equalization systems, this kind of serial equalizer
is considered as rather classical approach, and block transmission based equalization scheme
such as OFDM will be the main stream of the equalization scheme nowadays. Thus, more concrete
target of the wired line communications systems including specific standard should be clarified
in the paper instead of only "wired-line communication. Especially, it is hard to imagine a
wired-communications systems which have difficulty to use PAM4 in this era.

-The complexity of the proposed approach is compared as that of the FB algorithm to show the
effectiveness of the proposed approach, but it should be compared with frequency domain equalization
approach, which has much lower computational complexity of O(N log N).

-In the performance evaluation, the channel models used in the simulations will not be sufficient.
It is true that a certain pre-cursors can be recognized, but judging from Fig. 7, all channel models
are almost minimum phase channel. Please show the performance in much more non-minimum phase channel
as well.


**Summary Of The Paper:**

The paper considers the problem of channel equalization for wired-line communications systems,
and proposes a machine learning based equalizer inspired by forward-backward algorithm.

**Summary Of The Review:**

A certain contribution can be found in the proposed method, while a classical toy model is assumed. However, authors also claims the effectiveness of the proposed method in modern wire-line communications systems, which I cannot agree.

---

> ### Author Response · Authors · 2022-11-14
> **Respond to Reviewer X5rc**
>
> __W1__  From the view point of the development of equalization systems, this kind of serial equalizer is considered as rather classical approach, and block transmission based equalization scheme such as OFDM will be the main stream of the equalization scheme nowadays. Thus, more concrete target of the wired line communications systems including specific standard should be clarified in the paper instead of only "wired-line communication. Especially, it is hard to imagine a wired-communications systems which have difficulty to use PAM4 in this era.
>
> __W2__ The complexity of the proposed approach is compared as that of the FB algorithm to show the effectiveness of the proposed approach, but it should be compared with frequency domain equalization approach, which has much lower computational complexity of O(N log N).
>
> __Answer to W1 and W2__ Thank you for your kind remark. OFDM is a well-established, robust communication method, and frequency domain equalization is an effective approach. We agree that there weren't enough explanations about why focusing on serial equalizers when there were such alternatives. So we would like to explain why, in wireline communication, serial equalizers are favored over frequency domain equalizers.
> Until recently, NRZ (PAM2) was the main wireline transmission scheme due to simplicity and low power consumption. But increasing demand for higher data rates led to more severe channel loss at the Nyquist frequency. This caused the shift to PAM4. However, PAM4 with ISI has much more signal levels compared to NRZ, thus ADCs had to be added to the RX for more accurate signal detection using DSP(digital signal processing) such as FFE and DFE. Unfortunately, this significantly increased RX power consumption. On the other hand, it was possible to modify TX for PAM4 with relatively little increase in power consumption.
>
> But OFDM, despite many advantages, requires a substantial structural change in both the RX and TX. Especially, the need for IFFT and high resolution, high-speed DAC causes TX power consumption to be orders of magnitude greater. In contrast, NeuralEQ can be applied to the current high-speed PAM4 communication systems by only replacing the DSP engine on the RX side. It will be cheap and simple to apply our equalization method.
>
> In addition, many wireline communication specifications require backward compatibility. Unless all the users and companies decide to pay the huge cost for systemic changes in their wireline communication protocol, baseband transmission with pulse amplitude modulation will be the main wireline transmission scheme for a while.
>
> We can see this from how prospective ultra high-speed wireline communication specs(over 20Gbps)-PCIE 6.0, PCIE 7.0, IEEE 802.3ck, IEEE 802.3df, GDDR7-were all agreed to use a PAM4 system during recent discussions. More notably, there was an active discussion between DMT(discrete multitone transmission) and PAM4 in the IEEE 802.3bs ethernet(targets 200 or 400 Gbps) specification task force. Their final decision was also PAM4. Such history shows PAM4 and serial equalization will be dominant in wireline communication.
> With these circumstances, research on serial equalizers, for baseband transmission with pulse amplitude modulation, is crucial. We propose to tackle this problem with NeuralEQ.
>
> __W3__ In the performance evaluation, the channel models used in the simulations will not be sufficient. It is true that a certain pre-cursors can be recognized, but judging from Fig. 7, all channel models are almost minimum phase channel. Please show the performance in much more non-minimum phase channel as well.
>
> __Answer to W3__ Thank you for this comment. We realized we didn’t explain in detail how the tested channels in the paper were obtained. The channels were actually extracted from the real PCB striplines with different lengths. In addition, it is already a non-minimum phase channel with its zeros on the right half plane (in s-plane). More detailed information will be added to the paper soon.
> But we agree that testing NeuralEQ with more complex and non-minimum phase channels can provide more benefits of the proposed technique. We are currently testing our NeuralEQ on another channel and we’ll add the results as soon as possible.

---

### Author Response · Authors · 2022-11-18
**Paper is updated with supplments**

We thank the reviewer for their time and effort in reviewing the paper. The reviews have greatly helped to improve our paper's quality. In order to reflect your opinions, we have revised and re-uploaded the paper with some supplements as follows.
1. Additional channel simulation for NeuralEQ evaluation
2. Comparison of the number of flops for NeuralEQ and Forward-backward algorithm
3. Some typos and minor issues

In particular, we agree with the weakness that the types of channels used in the simulation are limited, and we have evaluated additional channels to alleviate this concern.
Thank you again, and it would be of great help to have additional comments.

---

### Decision · Program_Chairs · 2023-01-20

**Decision:**

Reject

**Justification For Why Not Higher Score:**

I cannot recommend acceptance of this paper because it does not appear to have significant contributions to the machine-learning community. As the authors state, the ICLR conference welcomes papers on applications. At the same time, as it is a competitive venue so that it would be desirable that papers presented there should be of interest for a wide audience. I judged that the contribution of this paper falls short of the bar in this respect.

**Justification For Why Not Lower Score:**

N/A

**Metareview: Summary, Strengths And Weaknesses:**

This paper proposes a neural-network-based equalizer for high-speed wireline communication. The proposed equalizer is inspired by the forward-backward (FB) algorithm for MAP inference in hidden Markov models, replacing building blocks in FB with fully-connected layers. It was demonstrated via computer simulations that the proposal provides better trade-off between BER and throughput than conventional approaches including FB as well as feed-forward equalizer (FFE) and/or decision-feedback equalizer (DFE). I share my concern on this paper with Reviewer Qiy5, that the proposal consists of replacing building blocks in a standard algorithm (FB) with trainable neural networks in a rather straightforward manner, and this idea itself could be regarded as falling within the category of deep unfolding and thus it is not novel enough from the perspective of deep learning.